# Telehealth Delivery of a Multi-Disciplinary Rehabilitation Programme for Upper Gastro-Intestinal Cancer: ReStOre@Home Feasibility Study

**DOI:** 10.3390/cancers14112707

**Published:** 2022-05-30

**Authors:** Louise Brennan, Fatemeh Sadeghi, Linda O’Neill, Emer Guinan, Laura Smyth, Grainne Sheill, Emily Smyth, Suzanne L. Doyle, Claire M. Timon, Deirdre Connolly, Jacintha O’Sullivan, John V. Reynolds, Juliette Hussey

**Affiliations:** 1Discipline of Physiotherapy, School of Medicine, Trinity College Dublin, D08 W9RT Dublin, Ireland; sadeghif@tcd.ie (F.S.); loneill4@tcd.ie (L.O.); laurahsmyth@gmail.com (L.S.); emythe3@tcd.ie (E.S.); jmhussey@tcd.ie (J.H.); 2Trinity St. James’s Cancer Institute, D08 NHY1 Dublin, Ireland; guinane1@tcd.ie (E.G.); sheillg@tcd.ie (G.S.); connoldm@tcd.ie (D.C.); osullij4@tcd.ie (J.O.); reynoldsjv@stjames.ie (J.V.R.); 3School of Medicine, Trinity College, D08 W9RT Dublin, Ireland; 4Physiotherapy Department, St. James Hospital, D08 NHY1 Dublin, Ireland; 5School of Biological and Health Sciences, Technological University Dublin, D07 ADY7 Dublin, Ireland; suzanne.doyle@tudublin.ie; 6Centre for eIntegrated Care, School of Nursing, Psychotherapy and Community Health, Dublin City University, D09 X984 Dublin, Ireland; claire.timon@dcu.ie; 7Discipline of Occupational Therapy, Trinity College, D08 W9RT Dublin, Ireland; 8Department of Surgery, Trinity Translational Medicine Institute, Trinity College Dublin, St James’s Hospital Dublin, D08 W9RT Dublin, Ireland

**Keywords:** survivorship, rehabilitation, exercise, nutrition, telehealth, feasibility, upper gastro-intestinal cancer

## Abstract

**Simple Summary:**

Throughout the COVID-19 pandemic, many cancer care services have safely been delivered via telehealth. Multi-disciplinary rehabilitation programmes can help address the complex physical, nutritional and quality of life needs of upper gastrointestinal (UGI) cancer survivors, but it is unknown how well these multi-component programmes translate to a telehealth model of delivery. Therefore, we assessed the feasibility of running a 12-week exercise and nutrition rehabilitation programme for UGI cancer via telehealth. Participants found the telehealth model safe, convenient and highly satisfactory. Lower levels of technology skills were a barrier to recruitment, and some participants needed help with using the technology. Some adaptations to how the exercise programme was delivered were required. Participants recommended that future versions of the programme would have some element of in-person contact. Cancer survivors should receive all possible supports to enable their participation in telehealth programmes.

**Abstract:**

Background: Telehealth has enabled access to rehabilitation throughout the pandemic. We assessed the feasibility of delivering a multi-disciplinary, multi-component rehabilitation programme (ReStOre@Home) to cancer survivors via telehealth. Methods: This single-arm mixed methods feasibility study recruited participants who had completed curative treatment for oesophago-gastric cancer for a 12-week telehealth rehabilitation programme, involving group resistance training, remotely monitored aerobic training, one-to-one dietetic counselling, one-to-one support calls and group education. The primary outcome was feasibility, measured by recruitment rates, attendance, retention, incidents, acceptability, Telehealth Usability Questionnaire (TUQ) and analysis of semi-structured interviews. Results: Characteristics of the twelve participants were: 65.42 ± 7.24 years; 11 male; 10.8 ± 3.9 months post-op; BMI 25.61 ± 4.37; received neoadjuvant chemotherapy 7/12; received adjuvant chemotherapy 4/12; hospital length of stay 16 days (median). Recruitment rate was 32.4%, and retention rate was 75%. Mean attendance was: education 90%; dietetics 90%; support calls 84%; resistance training 78%. Mean TUQ score was 4.69/5. Adaptations to the planned resistance training programme were required. Participants reported that ReStOre@Home enhanced physical and psychological wellbeing, and online delivery was convenient. Some reported a preference for in-person contact but felt that the online group sessions provided adequate peer support. Conclusion: Telehealth delivery of ReStOre@Home was most feasible in individuals with moderate to high levels of digital skills. Low level of digitals skills was a barrier to recruitment and retention. Participants reported high levels of programme adherence and participant satisfaction. Adaptations to future programmes, including introducing elements of in-person contact, are required.

## 1. Introduction

Advances in diagnosis and the treatment for upper gastro-intestinal (UGI) cancers have led to improved survival rates and, consequently, to a larger population of survivors of many types of UGI cancer [1,2]. Progress in survivorship care for UGI cancer remains poor, and many survivors experience ongoing negative physical and psychosocial impacts of treatment, which can have profound and long-term impacts on physical function and quality of life (QOL) [3,4]. At one year post-op, 40% of survivors report poor physical function, and significant reductions in walking distance, cardiorespiratory fitness and muscle strength are observed, along with a high prevalence of fatigue (41%), sarcopenia (35%) and dyspnoea (20%) [5,6,7]. Nutritional compromise in UGI cancer survivors is frequently reported, with eating restrictions are observed in 49% at 1 year post-surgery and malabsorption in 73% at two years post-op [6,8]. This can lead to significant reductions in fat-free body mass and skeletal muscle [8]. From a psychosocial perspective, anxiety (36%), fear of recurrence (29%) and high rates of sleep difficulties (51%) are reported. An integrated, multi-disciplinary specialist rehabilitation approach focusing on patient-centred outcomes is indicated to address the substantial, complex, multi-dimensional rehabilitation needs of UGI cancer survivors and to enable them to achieve the best possible quality of life and to reintegrate into family, social and working life [9,10,11,12].

The benefits of exercise in cancer care are well established [13,14,15]. There is strong evidence that aerobic and resistance exercise can improve cardiovascular fitness, physical function and QOL, and can reduce fatigue, pain, anxiety and depression [16,17]. Programmes with supervised and group-based exercise will benefit from greater adherence and improved rehabilitation outcomes [18,19]. The combination of exercise and dietetic counselling may enhance the individual effects of each treatment modality, resulting in greater improvements in muscle mass, function and QOL [11,12]. Group education which addresses a range in general cancer-specific survivorship issues, unmet needs from a multidisciplinary perspective, and which provides peer support is a third core component of multi-disciplinary rehabilitation [20,21,22,23]. There is a gap between recommendations and practice in UGI cancer survivorship care, and a multi-disciplinary rehabilitation approach is not usually formally applied as the standard of care [10,14].

The Rehabilitation Strategies for Oesophageal Cancer (ReStOre) multi-disciplinary programme combines group resistance and aerobic exercise training with one-to-one dietary consultations and group education sessions over a 12-week period, with the aim to improve cardiovascular fitness and health-related QOL [24]. A feasibility study and pilot randomised controlled trial (RCT) of the ReStOre programme found it to be feasible, safe, and valuable to patients and to be effective in improving cardiovascular fitness without compromising body composition [23,25,26,27,28]. A larger RCT is now planned (ReStOre II RCT) to evaluate the intervention across a wider cohort of people with upper gastrointestinal cancer [24].

The delivery of exercise and other cancer rehabilitation services has been seriously interrupted throughout the COVID-19 pandemic [29]. Exercising indoors, in groups, and particularly with vulnerable individuals, such as those recently undergoing cancer treatment or who are immunosuppressed, carries a high risk of viral transmission. Telehealth has emerged as an effective and safe way of delivering exercise programmes and other survivorship services during the pandemic [30,31,32]. It is now an increasingly routine method of service delivery due to its core benefits of convenience, improved efficiency, scalability and user satisfaction [33]. Telehealth can be defined as the delivery of healthcare services at a distance, using a range of information technology, including videocall, phone call, and text messaging, and can occur asynchronously or in real time [34,35]. Due to the variety of interventions classified as “telehealth”, it is challenging to compare the effectiveness and estimate the generalisability of telehealth programmes. In a UGI cancer population, exercise-only telehealth has been found to be feasible and achieved high rates of patient satisfaction [36]. However, the telehealth adoption of a multi-component, multi-disciplinary rehabilitation programme designed to address the complex needs of a cancer survivor population, such as ReStOre, is particularly challenging. An online model should aim to replicate all aspects of the programme, including group sessions, individual consultations, peer support and the social benefits of an in-person programme [23]. The effects of telehealth delivery on key feasibility outcomes such as adherence, intervention fidelity and patient satisfaction must be explored [37,38].

This study examined the feasibility of implementing ReStOre@Home, an online 12-week multidisciplinary rehabilitation programme consisting of aerobic and resistance exercise, dietetic counselling, and education sessions, which aims to improve physical fitness, nutritional status and quality of life in UGI cancer survivors. Feasibility was determined by recruitment rate, adherence rate, programme acceptability, retention rate and incidents. Secondary aims were to examine the effect of ReStOre@Home on physical function, dietary quality and nutritional status, fatigue and health-related QOL.

## 2. Materials and Methods

### 2.1. Study Design

This is a single-arm, mixed method feasibility study with a pre-test post-test design which is underpinned by the Medical Research Council (MRC) framework for evaluating complex interventions [37]. The study design is mapped to MRC framework to understand the feasibility of the intervention and to optimise its design and evaluation. This process aims to identify problems related to acceptability, compliance, delivery of the intervention (fidelity, adaptations, reach and dose), recruitment and retention [39], and is described in detail in O’Neill et al. (2021) [40]. A data management plan (extended data [40]) for ReStOre@Home describes data management during and after the intervention. Clinical Trial Registration Number: NCT04603339.

### 2.2. Participants and Recruitment

A recruitment target of *n* = 12 was based on sample size recommendations for pilot studies by Julious et al. [41]. Similar sample sizes are seen in comparable cancer rehabilitation pilot studies [42,43,44]. Inclusion criteria were:histological confirmed diagnosis of cancer of the oesophagus or stomach;≥three months post-oesophagectomy or total gastrectomy with curative intent;±neo-adjuvant/adjuvant chemo/chemoradiotherapy (completed) with curative intent;access to broadband internet;medical clearance to participate in intervention.

Exclusion criteria were: ongoing serious post-operative morbidity; evidence of active or recurrent disease; any serious co-morbidity that would impact exercise participation, including uncontrolled hypertension (resting systolic blood pressure >180 mmHg and/or diastolic >100 mmHg) or recent serious cardiovascular events (within 12 months) including but not limited to cerebrovascular accident, myocardial infarction, unstable or severe chronic disease (cardiac, renal, lung, liver or other), uncontrolled atrial fibrillation, and left ventricular function <50%. Fulfilment of exclusion criteria was determined by reviewing medical charts, safe completion of a medically supervised cardiopulmonary exercise test and consultations with participants and their medical team. All participants required medical clearance prior to enrolment.

Suitable participants were identified through eligibility screening of a patient database in St James’s Hospital, the National Centre for Oesophago-gastric Cancer in Ireland. We then sent a participant information leaflet and cover letter to eligible individuals and followed this up with a telephone call one week later. On receipt of verbal consent and medical clearance to participate, we scheduled an in-person baseline screening assessment in the Wellcome Trust HRB Clinical Research Facility at St James’s Hospital, Dublin, Ireland, where all participants provided written informed consent. Information on safety measures and COVID-19 risk reduction is detailed in the ReStOre@Home protocol38.

### 2.3. Intervention

#### 2.3.1. Intervention Details

ReStOre@Home is a 12-week telehealth exercise and nutrition rehabilitation programme for survivors of oesophago-gastric cancer. A programme overview is provided in Figure 1. ReStOre@Home was run via a Digital Therapeutics Platform created by Salaso Health Solutions Ltd. (Kerry, Ireland). The evidence-based and clinically-tested digital therapies platform allowed us to host videocalls (one-to-one and group), and provide exercise pre-scription and appointment scheduling, which enabled a reliable and user-friendly delivery of telehealth services.

The programme consists of the following core components:Aerobic and resistance exercise. Aerobic training was a walking programme of increasing frequency, duration and intensity (Figure 2), with heart rate targets individually prescribed using heart rate reserve (HRR), which was calculated as follows: 220—age = Max HR, Max HR—resting HR = HRR. Participants used Polar M200 heart rate monitor watches and the Polar Flow smartphone application (app) (Polar Electro Oy, Kempele, Finland). Researchers could remotely monitor walking data (duration, distance, speed, max heart rate, average heart rate) through shared access to the Polar Flow app. This information was used to set individualised goals with participants and to inform progressions of the walking plan. All equipment required for the exercise intervention was provided by the research team. Participants attended online supervised group resistance training sessions with a physiotherapist twice weekly. The resistance training programme used in these sessions (Figure 2) was developed for the ReStOre II randomised controlled trial [24] and was based upon a knowledge of participants’ 1-repetition maximum (1-RM) for each movement. From week 5, there was a structured, gradual transition from supervised to independent training (Figure 2), which was designed to promote self-management and long-term engagement exercise.Individual dietetic counselling. A registered dietician (FS) provided one-to-one dietary advice, education and goal setting. The dietetic counselling aimed to improve self-management of gastrointestinal symptoms and ensure adequate protein and energy intake [12].Multi-disciplinary education. These group sessions, aimed at addressing unmet information needs in UGI cancer survivorship, were provided by specialists from the UGI multi-disciplinary team. The topics covered were: introduction to group and programme, goal setting, physical activity, nutrition, fatigue, sleep and mental wellbeing. The information content was based upon participant needs, as identified at assessment and by consulting participants throughout the programme. Sessions focused on providing short bursts of information and then facilitating group discussion around the chosen topic. Education sessions were scheduled after the exercise class and lasted one hour.

Additionally, regular one-to-one check-in videocalls with a physiotherapist were scheduled. These involved support with exercise intervention, individualised goal setting and technological support. Each participant received a logbook to chart their weekly exercise (aerobic and resistance) and programme-related goals. The programme was scheduled as in Figure 1, and the frequency of dietetic counselling and check-in calls could be altered in line with participants’ individual needs. At T0 assessment, we provided participants with all equipment needed for the programme: dumbbells in a range of suitable weights, a heart rate monitor watch, tablet computer (where needed) and an information guide with advice on exercising, setting up the watches and telehealth software, and a contact number for the telehealth company who provided assistance directly to participants.

#### 2.3.2. Social Cognitive Theory

The design of ReStOre@Home is grounded in social cognitive theory (SCT), which posits that a person’s perceived self-efficacy influences health behaviours directly and also indirectly, through the impact of self-efficacy on goals, outcome expectation and perception of facilitators and barriers [45,46]. The core determinants of the theory which interact to produce changes in health behaviours are: perceived self-efficacy, knowledge of health risks, outcome expectations, health goals, perceived facilitators and socio-structural impediments. The design of ReStOre@Home addresses these core determinants as shown in Figure 3. Each component of the programme was designed to improve health knowledge and self-efficacy and therefore elicit a positive direct and indirect influence on health behaviours. Through the application of SCT, ReStOre@Home aims to enable participants with a greater self-efficacy over their recovery and to achieve healthy, lasting, lifestyle changes.

## 3. Outcomes

### 3.1. Outcome Schedule

Pre-intervention assessments (T0) were one week before the intervention commenced, and post-intervention assessments (T1) occurred one week after the intervention finished. All assessment outcomes were taken at both T0 and T1, apart from the TUQ and the qualitative interview, which occurred at T1 only.

### 3.2. Feasibility

#### 3.2.1. Quantitative Measures

The primary outcome of this study was feasibility. This was determined through the measures listed below. Where applicable, associated target outcomes for feasibility measure are listed. These were presented in the study protocol and were chosen based on findings of the in-person RestOre feasibility programme [25,40].

Recruitment rate: the percentage of eligible study population who consent to participation. Target: ≥50% of eligible patients recruited.Retention rate: the percentage of enrolled participants completing the post-intervention assessment. Target: ≥83% retention rate.Adherence: the total number of sessions attended for each programme component, total number of compliant exercise sessions, exercise dose modification and treatment interruption. Adherence was recorded using data from attendance logs kept by researchers, polar flow data, participant logbooks. Target: mean ≥80% adherence to supervised exercise sessions, ≥70% adherence to unsupervised sessions.Programme acceptability: determined through analysis of post-intervention interview and the Telehealth Usability Questionnaire findings.Incidents: defined as any unintended or unexpected incident that resulted in or could have resulted in harm to one or more patients. Incidents were reported by participants or researchers and recorded in case report forms and their implications on ReStOre@Home feasibility would be individually and carefully considered.

Data were also collected on participants’ readiness and ability to adopt and operate the various technologies required for ReStOre@Home. Data from the following sources were used to calculate adherence: researchers’ records of attended sessions, polar flow data, participant logbooks. The Telehealth Usability Questionnaire (TUQ) was chosen to assess participant satisfaction and intervention acceptability [47]. The TUQ is a 21-item questionnaire designed to evaluate computer-based, person-to-person healthcare interactions. It was scored on a 5-point Likert scale, with responses ranging from Completely Disagree (1) to Fully Agree (5). As stated in the study protocol [38], determining the feasibility of ReStOre@Home was guided by considering all feasibility findings and by specifically meeting the following targets: ≥50% of eligible patients recruited; mean of ≥80% adherence to supervised exercise sessions and ≥70% adherence to unsupervised sessions; ≥83% attendance at T1 assessment.

#### 3.2.2. Qualitative Interviews

Semi-structured interviews were held via videocall or in-person with all participants at T1 by members of the research team who were not involved in delivering the intervention. Standards for Reporting Qualitative Research (SRQR) guidelines are adhered to in the reporting of this qualitative work [48]. Researchers used an interview guide (Appendix A) covering all facets of the ReStOre@Home programme, including impact on fitness, nutrition and daily life; participant preferences and recommendations for future programmes; and experiences with the technology. Interviews were audio-recorded and then transcribed verbatim. Two researchers (LB and LON) conducted reflexive thematic analysis with this data, following the process outlined by Braun and Clarke [49], applying a deductive approach and coding for topics related to feasibility of telehealth delivery of the programme. Full analysis of all topics discussed will be presented in a separate manuscript. Final themes were then agreed, defined, and named by both researchers. For clarity of reporting: both researchers were physiotherapists and experienced in qualitative data analysis; LB was involved in intervention delivery, and LON conducted the interviews.

#### 3.2.3. Secondary Outcomes and Biobank

Secondary aims were to examine the effect of the ReStOre@Home programme on physical functioning, body composition, dietary intake, nutritional status and patient-reported outcomes, including health-related QOL and fatigue. Secondary measures were taken at T0 and T1 and consisted of: physical function: cardiopulmonary exercise test (protocol described in O’Neill 2020 [24]), short physical performance battery [50], hand grip strength [51], leg press 1-RM [24]; physical activity levels: Godin-Shephard Leisure Time Physical Activity Questionnaire [52]; body composition: anthropometry, mid-arm and waist circumference, bioimpedance analysis; dietary intake: dietary interview, 24 h dietary recall (FoodBook24) [53]; nutrition-related symptoms: Gastrointestinal Symptom Rating Scale (GSRS) [54]; Simplified Nutritional Appetite Questionnaire (SNAQ) [55]; QOL: EORTC-QLQ-C30 [49], EORTC-QLQ-OG25 (0esophago-gastric cancer); Fatigue: Multidimensional Fatigue Inventory (MFI-20) [56]; participant experience and feedback: semi-structured interviews. Secondary measures were exploratory only, as the sample size is not sufficient to demonstrate treatment effect.

Participants were invited to consent to donate a blood sample to the UGI Survivorship Biobank, which was established alongside the ReStOre II RCT [24]. Samples were gathered at T0 and T1, processed and stored at −80 °C at the Trinity Translational Medicine Institute, St James’s Hospital, Dublin 8.

## 4. Results

### 4.1. Recruitment and Retention

A flow chart of recruitment and retention is presented in Figure 4. Forty-eight received the invitation letter and participant information leaflet, and the full recruitment process (invitation and follow up phone call) was completed with 37 people, as the target for recruitment (12) was then met. Recruitment rate was 32.4%. The online model was the main reason for declining participation for 22% (8/37) of potential participant. Retention rate was 75%: three left the programme in the second week, due to illness (*n* = 1), difficulties with the technologies which could not be overcome with support from researchers or family (*n* = 1), and challenges related to an intense work schedule (*n* = 1). There were no adverse incidents throughout the 12-week study.

### 4.2. Participant Characteristics and Technological Abilities

Participants were mainly male (*n* = 11), with an average age of 65 years and a history of oesophageal cancer. Participant characteristics at baseline and ability to use the technologies in ReStOre@Home are presented in Table 1. We provided tablets to two participants who did not have a suitable device at home. One participant did not have broadband and used the internet in his daughter’s house. Three participants had support from family members to join the calls, as they were not habitual internet users.

### 4.3. Adherence

#### 4.3.1. Attendance

Data results presented from here on are from participants who completed the programme (*n* = 9). Attendance is presented in Table 2.

#### 4.3.2. Fidelity and Dose

The programme schedule and the group education sessions were delivered as planned (Figure 1). The planned aerobic training programme was adapted for two participants, one of whom experienced a fall at home and one who had an exacerbation of low back pain. These participants were medically cleared to perform walking-based exercise, and their walking programmes were adapted to short times and lower intensities. It was not possible to closely adhere to the planned intensity of the resistance training programme (Figure 2), as calculating 1-RM for individual participants via telehealth using a limited selection of weights was challenging. The programme delivered involved progressions in load, repetitions, and sets over the 12 weeks; these metrics were charted for each participant at each session by ReStOre@Home physiotherapists. Mid-way through the programme, heavier weights were sent to participants who required them. As a result of the change in resistance training delivery, the adherence measures “total number of compliant resistance sessions” and “dose modification” could not be calculated. Due to problems in reliably collecting heart rate data from four participants, the measure “total number of compliant aerobic sessions” was not completed.

Treatment interruption in supervised resistance training occurred in two participants, due to illness (*n* = 1, 3 sessions missed) and return to work (*n* = 1, 9 sessions missed but all completed independently, unsupervised later in the day). No sessions were terminated early.

#### 4.3.3. Telehealth Usability Questionnaire

Results from the TUQ are presented in Table 3.

#### 4.3.4. Achievement of Feasibility Targets

The feasibility targets and observed rates are presented in Table 4.

#### 4.3.5. Secondary Measures

Secondary measures are presented as Appendix A: Physical outcomes, leisure time activity, anthropometric measures and bioimpedance analysis; Appendix A: QOL outcomes; Appendix A: Nutrition-related symptoms.

### 4.4. Qualitative Feasibility Data

Three main themes relating to feasibility of the telehealth model of ReStOre@Home were noted: participant experiences with telehealth technology; benefits to patients of the telehealth model; and challenges of the telehealth model and recommendations.

#### 4.4.1. Participant Experiences with Telehealth Technology

While participants with experience of telehealth were typically confident about their ability to take part in the programme, participants with limited experience with technology were more apprehensive:


*“When I heard it first, I thought I’m not au-fait with tech that much and I was kind of nervous.”*
P12

Joining sessions on the telehealth platform was a user-friendly process, and most participants could operate the platform with ease and independently:


*“I don’t think it could be much easier, I mean you just…click “join call” and that’s it. Only two clicks and you’re there. I couldn’t believe it was so easy.”*
P12

The heart rate monitor watches were an effective support for some participants:


*“[The physiotherapists] set me goals with the Polar watch, and I found that I reacted to that and bought into it, and I felt better.”*
P6

However, others struggled with the watches, and two participants eventually abandoned the devices:


*“I haven’t used it. I said I have to give it up. It didn’t last long, it was going dead too quick. I mean, I didn’t fully understand it.”*
P10

Family members provided important support in accessing the technology. In some cases, the support was minimal, as P7 explained: “once [my wife] showed me the way, I was good”. In other cases, assistance was needed throughout the programme. However, this was not perceived as a major barrier:


*“I had no bother, my daughter here set up everything. I haven’t got Wifi or anything at home. But that wasn’t much of an inconvenience.”*
P5

#### 4.4.2. Benefits to Patients of the Telehealth Model

All participants found the telehealth aspect of the programme beneficial and convenient:


*“The unique thing about this is its remote, possibly makes it even more feasible, and more doable for the patient you know?”*
P1

Reduced travel was a considerable advantage of telehealth, as expressed by P7: “The mere fact you don’t have to travel is brilliant.” This relieved pressure on participants who depended on family members for transport:


*“It’s really handy, as in, you don’t have the bother of trying to get somebody to bring you there. You’re not relying on them to come and collect you. That’s a good thing.”*
P2

The ability to join sessions from any location with an internet connection was appreciated and enabled participants living far from the hospital to attend. Additionally, others attended during work breaks, or when out of the country. Through this, participants could make the programme fit around their lives with minimal disruption:


*“I was able to do this from Dublin and Spain. I hadn’t been away for two and a half years. We booked in hope to come to Spain way before I knew about the programme.”*
P11

Participants appreciated the safe and convenient option of telehealth during the ongoing COVID-19 pandemic. P1 explained: “There are huge benefits in that it reduces cross contamination in covid times”. Although participants could not meet in-person, the group aspect of ReStOre@Home allowed participants to learn from peers and generated a feeling of community:


*“It gives you a sense that you belonged to something. And that’s one thing I found that I’ve missed. It helps you to integrate with other people.”*
P2

#### 4.4.3. Challenges of the Telehealth Model and Recommendations

The telehealth software and heart rate monitor watch were new technologies, and some recommended that the process of getting set up with the technology could be simplified:


*“Possibly a couple of hours that everybody could come into the hospital and you could advise some of the administration, some of the technology, how it works.”*
P11

Several participants wished to keep the equipment used in the programme to allow them to continue with the home-based programme independently. P6 described this as “one of the downsides…handing back the stuff, I’d like to keep it all.”

While the telehealth model was well-accepted, participants still placed a high value on meeting in-person:


*“I suppose, the ideal thing if you could have a class in a room with people, the physical presence is nicer and better.”*
P12

They suggested that future iterations of the programme could adopt a hybrid model:


*“I think they could actually do it, even as a hybrid thing, get people in maybe once a month and the rest could be done on Zoom, or whatever”*
P9

An alternative suggestion was to run concurrent in-person and online sessions, to provide flexibility and choice for patients:


*“Allowing people to continue not to come physically and attend remotely, even though some people are physically there, would be a good thing”*
P1

## 5. Discussion

In this study, we assessed the feasibility of delivering a multi-disciplinary, multi-component rehabilitation programme designed for a cancer survivor population with complex needs. To the best of our knowledge, this is the first study of a telehealth multimodal rehabilitation programme in UGI cancer. Recruitment and retention rates were lower than the in-person version of ReStOre, and attendance was high for most programme components [25]. High levels of participant satisfaction and acceptability were seen in the TUQ and interview findings. The remote nature of ReStOre@Home was convenient; however, it was important to participants that future versions of the programme would have some aspect of in-person contact. ReStOre@Home enabled a vulnerable cohort of people to take part remotely in group exercise and rehabilitation, including groups with substantial barriers to attendance, such as those living far from the specialist cancer centre and those who had returned to work. The feasibility findings were mixed overall: ReStOre@Home was feasible in people with the skills and resources to use the technology; however, it was less feasible for people with lower levels of digital skills.

The recruitment rate of 32% was lower than our target of ≥50%, which was based upon the ReStOre in-person recruitment rate of 55% [25]. However, there are distinct challenges for recruitment to telehealth [57], and recruitment rates in telehealth interventions for cancer populations vary widely, depending on the intervention, population and recruitment practices [58,59,60,61]. High refusal and attrition rates have been observed in other multi-component rehabilitation interventions in cancer care [62]. There are two factors which strongly contributed to the lower recruitment rates in ReStOre@Home. First, seven potential participants were not in a suitable phase of recovery to engage in the programme (recent illness preventing participation *n* = 3, self-report “not ready” *n* = 2, intervention not needed *n* = 2). Second, a large proportion of patients who declined (8/15) were not interested in the online model. It can be expected that an online intervention will not be suitable for a certain proportion of the population in Ireland, as 16% of homes do not have a fixed broadband connection [63] and adult digital literacy levels are below the European Union average [64]. In-person services should be available for those who cannot take part in online programmes [65].

The retention rate of 75% was slightly lower than the target of ≥83%. One participant left the trial, as he had difficulties with the technology, which could not be resolved with support from researchers and family. To improve recruitment and retention rates, it is evidently important to understand participants’ digital readiness, and this could be identified through a measure such as the Readiness and Enablement Index for Health Technology, which has been validated in a cancer rehabilitation setting [66]. Further tactics to improve recruitment and retention for multi-disciplinary telehealth rehabilitation programmes include providing clear and careful messaging about the trial, improving accessibility of participant information leaflets and increasing in-person methods of recruitment [67,68]. Using the sample of 37 people who were approached for this study, we estimate that a maximum of 70% could theoretically be suitable to engage in an online intervention (calculated by excluding seven non-respondents, eight who declined due to the online model, and one who left the study due to technology issues). This is a rough estimate made from a small sample, but the figure may contribute towards generating approximate targets for recruitment in future trials.

High levels of participant satisfaction and acceptability were observed in TUQ results, particularly in the subscales of usefulness (4.96/5) and interaction quality (4.83/5). Perceived usefulness is an important predictor of engagement in digital health technology [69]. These findings indicate that the telehealth model was convenient, improved access to healthcare and delivered high-quality healthcare interactions. Notably, the lowest-scoring TUQ statement was, “I think the visits provided over the telehealth system are the same as in-person visits” (4.11/5), suggesting that an important aspect of in-person healthcare was not emulated by the telehealth system. This sentiment is echoed throughout the literature on telehealth; patients often perceive that telehealth interventions lack the “personal touch” [70].

The qualitative data present some mixed findings regarding participant satisfaction. Participants appreciated the convenience of the telehealth model, which reduced travel time and facilitated attending the programme alongside participants other commitments and activities. Eliminating travel-related barriers to rehabilitation is a fundamental strength of the telehealth approach [71]. In the in-person model of ReStOre, a primary reason for declining participation was travel being too arduous [26]. People with prohibitively long travel times, with physical limitations to travel, and without social supports can more easily engage in telerehabilitation from their homes. However, some participants reported difficulties with the technology, and there was a desire for in-person contacts. Future iterations of ReStOre could address this by using a hybrid model of service delivery to provide greater patient choice and to potentially increase recruitment [72,73].

Attendance rates were high for all programme components apart from supervised exercise, which was 78% (unsupervised was 85%). This is similar to findings of the in-person ReStOre feasibility trial (supervised 82%; unsupervised 78%), and a multi-component cancer rehabilitation programme conducted by Dennett et al. (2021), [74] which had 80% attendance to one-to-one telehealth sessions.

The MRC framework for evaluating complex interventions recommends that feasibility studies examine intervention fidelity and dose and the influence of intervention context [37]. Through pre-intervention assessments and remote heart rate monitoring, we were able to maintain high fidelity to the aerobic training programme and deliver the planned dose for the majority of participants. However, the online context negatively impacted fidelity and dose in the resistance-training programme. It was not possible to perform a 1-RM testing remotely, as a range of different, often heavy, weights are required for this, and some participants would need in-person support if they were unfamiliar with weight training. A balance of safety and intervention fidelity is required when converting rehabilitation interventions to a telehealth model [75]. Therefore, participants used a load which was perceived to be challenging for the number of repetitions in the session, and as a result, the load and dose used differed to that in the prescription detailed in Figure 2. To overcome the challenges to strength training via telehealth, other studies have applied time-based sets and have monitored exertion during training [76,77]. Future feasibility work could explore approaches which allow for more accurate prescription and monitoring, such as repetitions in reserve or using 1-RM prediction calculations [78,79]. Other internet-based exercise programmes did not use live telehealth group calls, and instead provided thorough instructions for exercises, delivered either in-person or online, which patients would perform unsupervised at home [80,81].

Reflecting on the findings of this study, we observed that translating or developing multi-disciplinary, multi-component cancer rehabilitation programmes for telehealth delivery is worthwhile and impactful, but requires careful, iterative planning to be successful. Our key recommendations for those developing similar programmmes are detailed in this paragraph. First, elements of in-person contact should be included to increase participant engagement and enhance peer support where possible. This could be through occasional in-person sessions or a hybrid model of delivery. Prior to commencing the programme, an in-person education session about the programme’s technologies could help reduce technology-related difficulties which may impact attendance and remote monitoring capabilities. Regarding digital literacy, we recommend that a simple screening of a participant’s level of digital skills, digital resources and support resources is conducted prior to starting the study. Additional supports should be made available to include all who wish to participate, and an in-person alternative of equal quality should be available for those who cannot attend remotely. Lastly, we recommend that programmes invest in reliable, user-friendly telehealth software, as this contributed greatly to the high participant satisfaction observed in ReStOre@Home and our ability to deliver the programme completely as scheduled.

### Limitations and Strengths

The small sample size in this study was chosen to be suitable for the primary outcome (feasibility) but was not sufficient to enable meaningful statistical analysis of secondary outcomes. As a result, it is difficult to generalise the findings of the secondary measures in this study. The high proportion of males in the sample may have influenced aspects of the programme, for example, the nature of the group discussions or the outcomes. The ReStOre programme is not suitable for individuals with low levels of physical function, who may need one-to-one support to engage in physical activity. These feasibility findings therefore do not apply to this group, and it is important that alternative models of rehabilitation are developed for those with high physical rehabilitation needs. The single-arm design did not allow for comparison with a control group; however, we can compare some outcomes to the in-person ReStOre feasibility study [25]. Strengths of this study were the broad recruitment criteria which resulted in diversity across age, level of impairment and technology skills, and the use of a telehealth platform which is specially designed for rehabilitation settings. Further strengths were the application of the MRC framework to ensure that a thorough feasibility evaluation occurred and the application of a mixed-methods approach, which utilised a wide range of quantitative and qualitative data to enhance the feasibility findings and include participants’ perspectives.

## 6. Conclusions

Multi-component, multi-disciplinary rehabilitation programmes are complex interventions, and little is known about how well they translate to a telehealth model of delivery. In this study, we evaluated the feasibility of a 12-week multi-disciplinary telehealth rehabilitation programme for UGI cancer (a telehealth model of the in-person ReStOre programme). Feasibility findings presented a mixed picture: the programme was more feasible in patients with moderate to high levels of technology skills, and low levels of digital skills was a barrier to recruitment and retention. Adaptations to the resistance training programme were needed to enable remote delivery. High levels of satisfaction and acceptability were noted among participants who completed the study. For participants, the telehealth model was convenient, safe and eliminated geographical barriers to access. Further work is required to translate multi-disciplinary rehabilitation programmes to telehealth delivery in a manner that is feasible for a broad patient population. To improve reach and retention, future programmes should incorporate some in-person sessions and explore how to improve the feasibility of telehealth for those with poor digital skills.

## Figures and Tables

**Figure 1 cancers-14-02707-f001:**
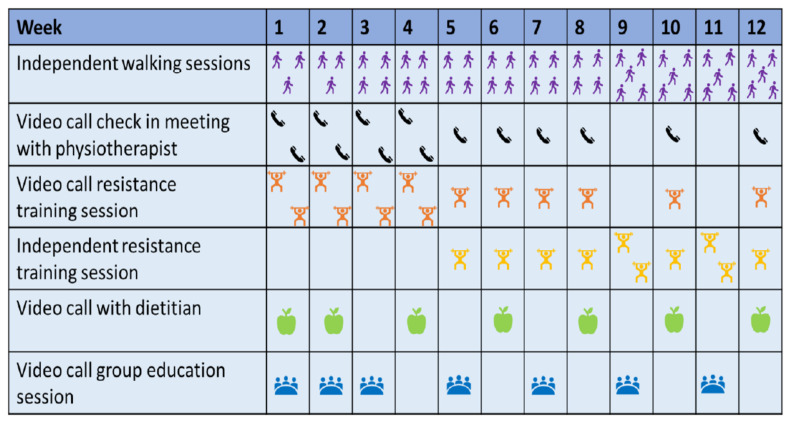
RestOre@Home 12-week programme schedule.

**Figure 2 cancers-14-02707-f002:**
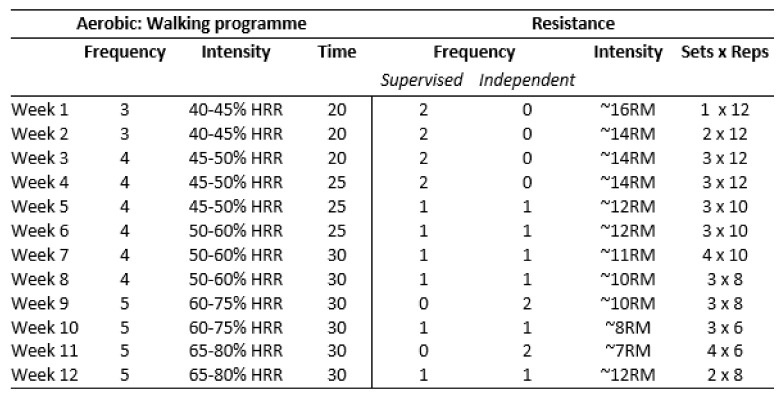
Exercise programme for ReStOre@Home, as adapted from ReStOre II randomised controlled trial protocol. HRR: heart rate reserve; RM: repetition maximum.

**Figure 3 cancers-14-02707-f003:**
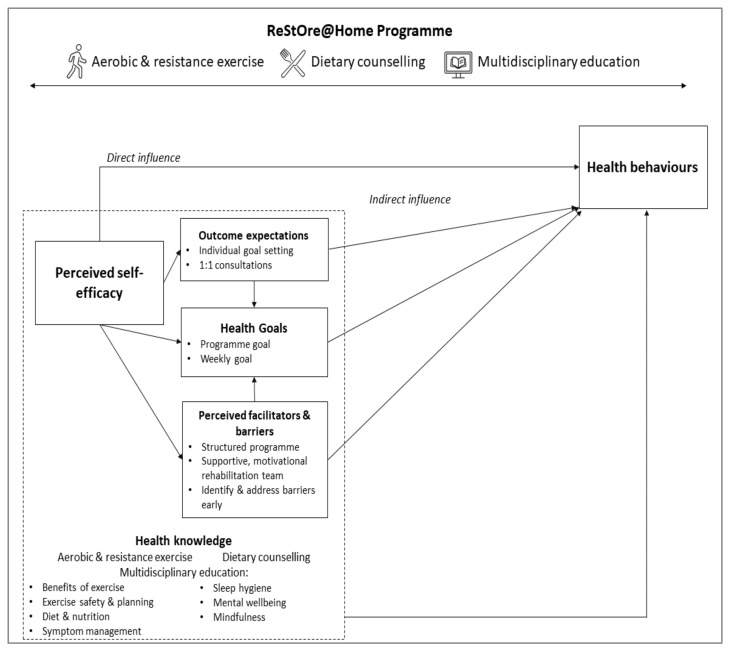
Social cognitive theory underpinning ReStOre@Home programme design, demonstrating the influence of health knowledge on the core determinants, and the interactions between determinants which result in changes in health behaviour [46].

**Figure 4 cancers-14-02707-f004:**
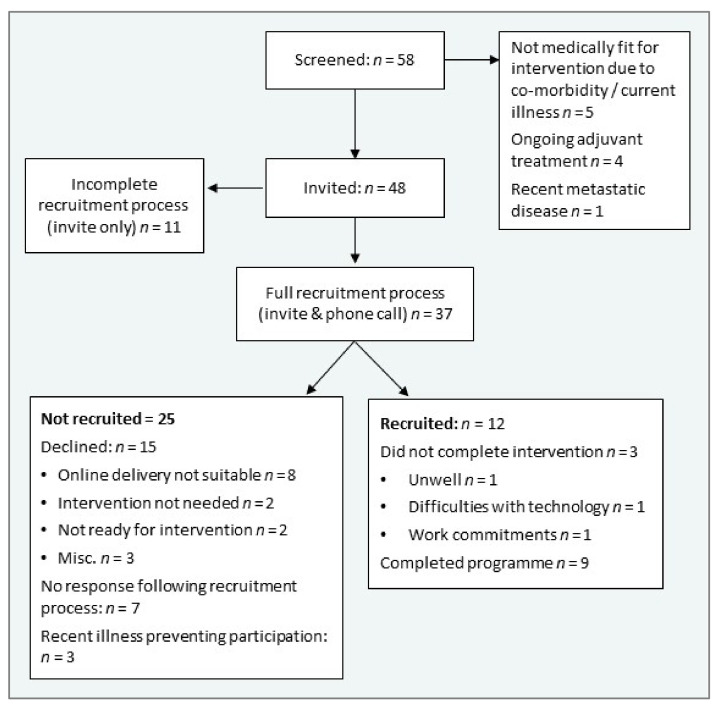
Recruitment and retention.

**Table 1 cancers-14-02707-t001:** Participant characteristics at baseline (*n* = 12) and ability to utilise the technologies used in ReStOre@Home.

Characteristic		Value	
**Age (years), mean (SD; range)**		65.42 (7.24; 53–76)	
**Sex, *n* (%)**	Male	11 (92%)	
	Female	1 (8%)	
**BMI, mean (SD, range)**		25.61 (4.32; 17.9–33.1)	
**Cancer type**	Oesophageal	10 (83%)	
	Gastric and lung	1 (8%)	
	Oesophago-gastric junction	1 (8%)	
**Time since surgery (months)**		10.8 (3.9; 5–17)	
**Neoadjuvant treatment (yes/no)**		7/5	
**Adjuvant treatment (yes/no)**		4/8	
**Hospital length of stay (days), median (IQR)**		16 (15)	
**Technology factor**		Yes (*n*/12)	No (*n*/12)
**Broadband internet access in own home**		11	1
**Access to suitable device for videocalls**		10	2
**Independently operated videocalls**		9	3
**Independently operated watch**		9	3

**Table 2 cancers-14-02707-t002:** Attendance to different programme components. Mean and SD relate to each individual participant’s overall attendance at each component.

Programme Component	Session Attendance	
	Mean Number Attended/Total Number (Range)	Percentage (SD)
**Education**	6.33/7 (4–7)	90 (15.9)
**Resistance Training supervised ***	10.89/14 (5–14)	78 (20)
**Resistance Training unsupervised**	8.56/10 (2–10)	85 (27)
	**Percentage of scheduled calls attended (SD) ****	
**Physiotherapy check-in calls**	84 (14)	
**Dietetic calls**	90 (14)	

* One participant returned to work at week 4 and completed all subsequent sessions unsupervised; mean adherence rate excluding this participant is 84% (SD 12). ** Mean number not provided, as number of scheduled calls differed per participant as per their needs: PT calls range from 11–14, Dietetic calls range from 4–7. SD: standard deviation.

**Table 3 cancers-14-02707-t003:** Telehealth Usability Questionnaire.

Item and Subscale	Mean Score (SD), Max = 5
**Overall Usability (all questions)**	**4.69 (0.24)**
**Usefulness**	4.96 (0.2)
**1. Telehealth improves my access to healthcare services**	5 (0)
**2. Telehealth saves me time traveling to a hospital or specialist clinic**	5 (0)
**3. Telehealth provides for my healthcare needs**	4.88 (0.35)
**Ease of Use and Learnability**	**4.37 (0.17)**
**1. It was simple to use this system**	4.33 (1.0)
**2. It was easy to learn to use this system**	4.56 (1.01)
**3. I believe I could become productive quickly using this system**	4.22 (0.97)
**Interface quality**	**4.72 (0.14)**
**1. The way I interact with this system is pleasant**	4.89 (0.33)
**2. I like using the system**	4.67 (0.71)
**3. The system is simple and easy to understand**	4.78 (0.44)
**4. This system is able to do everything I would want it to be able to do**	4.56 (0.53)
**Interaction quality**	**4.83 (0.06)**
**1. I could easily talk to the clinician using the telehealth system**	4.89 (0.33)
**2. I could hear the clinician clearly using the telehealth system**	4.78 (0.67)
**3. I felt I was able to express myself effectively**	4.89 (0.33)
**4. Using the telehealth system, I could see the clinician as well as if we met in person**	4.78 (0.44)
**Reliability**	**4.41 (0.28)**
**1. I think the visits provided over the telehealth system are the same as in-person visits**	4.11 (1.36)
**2. Whenever I made a mistake using the system, I could recover easily and quickly**	4.44 (0.73)
**3. The system gave error messages that clearly told me how to fix problems.**	4.67 (0.71)
**Satisfaction and future use**	**4.83 (0.06)**
**1. I feel comfortable communicating with the clinician using the telehealth system.**	4.78 (0.44)
**2. Telehealth is an acceptable way to receive healthcare services**	4.78 (0.44)
**3. I would use telehealth services again**	4.89 (0.33)
**4. Overall, I am satisfied with this telehealth system**	4.89 (0.33)

**Table 4 cancers-14-02707-t004:** Achievement of feasibility targets set at study design stage.

Measure	Target Rate (%)	Observed Rate (%)
**Recruitment of eligible participants**	≥50	32
**Adherence to supervised exercise sessions ***	≥80	78
**Adherence to unsupervised exercise sessions ****	≥70	85
**Attendance at T1 assessment**	≥83	75

* Percentage of supervised sessions attended; ** Percentage of completed unsupervised sessions, as reported in participant logbooks and on check-in calls.

## Data Availability

The data presented in this study are available on request from the corresponding author.

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
