# Peer review of "Telehealth Delivery of a Multi-Disciplinary Rehabilitation Programme for Upper Gastro-Intestinal Cancer: ReStOre@Home Feasibility Study"

_cancers, 2022, doi:10.3390/cancers14112707_

Round 1

Reviewer 1 Report

Thanks for your invitation to review this article. Here, Dr. Brennan and her colleagues assessed the feasibility of a 12-week multi-disciplinary rehab program for UGI cancer patients via telehealth.  They found this proof-of-concept study is overall successful and had a few recommendations for future development. This program is well-designed and it has great potential for UGI cancer survivors or even a more general group of patients with other malignancies.  I would recommend accepting this manuscript in principle if authors make a few minor revisions.

  1. As the authors discussed, the recruitment rates were low due to low internet literacy/availability and patients’ health status. While I understand this is a rehab program, a modification plan of the exercise program may be vital for patients with suboptimal performance status. In fact, these patients may benefit more from this rehab program. For example, early interdisciplinary supportive care has been shown to improve overall status in metastatic UGI cancer patients, who may be sicker than the UGI cancer survivors. The authors may like to further elaborate on this.  
  2. Regarding the technical difficulty, I am worried that the teleheatlh may be challenging for patients who are less educated, poorer, and live in rural areas. I am curious if the authors have any plan to partner with local community leaders to increase their access to this program in future trials. For example, setting up computers and volunteers in local community centers may help patients with technical difficulties.
  3. Other major strengths of telehealth rehab programs that the authors missed to discuss in their manuscript is: 1) telehealth may overcome transportation barriers. This should be discussed in detail in the discussion. We knew that cancer survivors are more likely to have transportation barriers, especially those with functional limitations, comorbid diseases, and without social support. The telehealth program may be particularly helpful for this group of survivors. 2) telehealth is easily scalable and may help to narrow the disparity in access to high-quality rehab care.

  1. The authors missed a few important articles in this field. They may like to discuss this paper in the revision.

        1).Early Interdisciplinary Supportive Care in Patients With Previously Untreated Metastatic Esophagogastric Cancer: A Phase III Randomized Controlled Trial

Zhihao Lu, Yu Fang, Chang Liu, Xiaotian Zhang, Xiaowei Xin, Yi He, Yanshuo Cao, Xi Jiao, Tianqi Sun, Ying Pang, Yanli Wang, Jun Zhou, Changsong Qi, Jifang Gong, Xicheng Wang, Jian Li, Lili Tang, and Lin Shen

Journal of Clinical Oncology 2021 39:7, 748-756

  1. )Jiang C, Yabroff KR, Deng L, et al. Self-reported Transportation Barriers to Health Care Among US Cancer Survivors. JAMA Oncol. Published online March 24, 2022. doi:10.1001/jamaoncol.2022.0143

Reviewer 2 Report

This pilot study has investigated the feasibility of translating an established multi-disciplinary rehabilitation program for upper-GI Cancer survivors into a telehealth delivered program and has taken a mixed methods approach to address key quantitative feasibility criteria as well as a qualitative analysis of patient interviews after experiencing the delivery of the program.

This is a crucially important area of research that aims to investigate the feasibility of translating diet and exercise programs into virtual service delivery models, many of which have attempted rapid roll-out into clinical practice due to the covid-19 pandemic.

Main comments

Sample size was far too low to interpret any of the effectiveness data (for some n was as low as 9 participants and indeed the authors rightly did not perform any statistical analysis) and so I recommend all tables with effectiveness data (Tables 5-7) are pulled from the main manuscript and accessible as supplementary files only. It is distracting to have to wade through the clinical data which is not helpful in determining feasibility.  The sample size of 12 was chosen as ‘suitable’ for feasibility but there is no justification given for this?

Interpretation of Feasibility:

The primary outcome measure of feasibility measures defined in bullet points across lines 229 – 239 should have the targets/cut-off points included instead of described later in the text (line 250 -251). For example, it’s not clear how a target/cutoff was chosen to determine feasibility for Program acceptability as per TUQ and inclusion of the qualitative interviews as a criteria is problematic as this data can’t be quantitatively defined with cut-offs and so should be separated from the quantitative criteria. What was the definition of an ‘incident’ and how was this data collection ie self report by participant or chart review? And what was the acceptable level of incidents that would deem the program safe or unsafe?

Of the 4 feasibility criteria presented in Table 4, only x1 was met (and based on self-reported logs). In addition, x3 prospectively described feasibility criteria could not be measured and were therefore abandoned. Therefore, I think the overall conclusion that the program is broadly feasible does not align with the quantitative data presented as per your pre-defined feasibility criteria.

I could not easily see any incident data?

The qualitative data illustrated some contrasted experiences that would indicate the program is perceived by patients as highly acceptable and feasible in some but not others and this is an important finding.

However, based on the results presented, the overall conclusion must reflect that translation of diet and exercise programs into virtual models of care is complex and despite some patients deeming it an acceptable service model, there remains considerable aspects of feasibility that need to be addressed in the future design of telehealth service models.

There is a significant evidence gap in the literature to inform how best to develop remote monitoring / outcome measures for diet and exercise telehealth interventions and this has clearly had a significant impact on the ability to assess some components of feasibility. This is a conundrum facing everyone around the world and so it would strengthen the paper to explore this in the discussion more and would be of great interest to many reading your paper who are struggling with similar issues.

There is no mention of safety screening procedures prior to commencement – were these undertaken and if so please reference or describe.

Line 327: There needs to be greater description of what topics were covered in the group education and dietetic sessions. Were these topics patient led? How was the fidelity of the delivery of these groups measured? Did each participant get exposed to the same education topics?

Line 86: What does the delivery exercise rehabilitation service look like in usual cancer care? This could be added to the introduction to orientate the reader to what patients would normally be exposed to in your clinical practice.

Line 107: The introduction reads well but could benefit from acknowledging that ‘telehealth’ is a broad umbrella term for any number of type and combinations of virtual/digital health care delivery including telephone, text messaging, videoconferencing, synchronous v asynchronous telehealth care. This makes critiquing effectiveness in the literature difficult and is reflective of the emerging nature of digital health care transformation.

Line 161:  How do you determine maximum heart rate for this equation (CPET)? The reference given only states the heart rate reserve intensity zones (Low: 40-45%HRR; Mod: 65-85%HRR). Within that paper there is reference to another protocol which states both resting and maximal heart rates from the CPET but is that what you did? In short, your methodology should be made more clear for the reader.

Figure 2 and discussion: RM intensity monitoring is clearly difficult/impossible over telehealth due to the need for equipment and capability of videoconferencing software. Could the authors expand in the discussion how clinicians and researcher in the field will need to innovate and potentially abandon traditional outcome measures and suggest potential solutions eg could a better option be repetitions in reserve (RiR) as they are related to intensity markers (ie. 2-4 RiR = mod. Intensity etc.) and generally easier for patient to understand/follow?

Line 254: Interview questions need to be included as a supplementary file or appendix

Weaknesses

Within the discussion there must be a paragraph describing study weaknesses and include an acknowledgement of the gender imbalance (11 out of 12 were male) which can impact group education dynamics and/or translatability of results. In addition the exclusion criteria seems to exclude many chronic diseases that might be targets for diet and exercise intervention and without a table describing the presence (or absence) of co-morbidities, it’s not clear if this patient group were otherwise quite well (albeit survivors of recent cancer treatment). This also relates to translatability into real world practice where the presence of obesity, metabolic syndrome and depression are very common factors in people seeking or being prescribed diet and exercise programs and often associated with co-morbidity ‘incidents’ that might impact the adherence to lifestyle programs. It’s stated on line 453 that the patient group has complex needs but I'm not sure there is data presented to support what those complex needs are? The presence of multimorbidity is a significant barrier to translating virtual lifestyle interventions and so it would be worthwhile including mention of this in your discussion, especially if there were no incidents reported which may reflect a safe program, but may also reflect a relatively well group of participants.

Round 2

Reviewer 2 Report

Authors have done an excellent job addressing all the concerns raised.

Author Response

Thank you for your comments and feedback to this manuscript.